# BanLemma: A Word Formation Dependent Rule and Dictionary Based Bangla Lemmatizer

**Sadia Afrin**[1]**, Md. Shahad Mahmud Chowdhury**[1]**, Md. Ekramul Islam**[1]**,**
**Faisal Ahamed Khan**[1]*****, Labib Imam Chowdhury**[1]**, MD. Motahar Mahtab**[1]**,**
**Nazifa Nuha Chowdhury**[1]**, Massud Forkan**[1]**, Neelima Kundu**[1]**, Hakim Arif**[2]**,**
**Mohammad Mamun Or Rashid**[3]**, Mohammad Ruhul Amin**[4]**, Nabeel Mohammed**[5]

[1]Giga Tech Limited, Dhaka, Bangladesh, [2]University of Dhaka, Bangladesh
[3]Bangladesh Computer Council, Dhaka, Bangladesh, [4]Fordham University, New York, USA,
[5]North South University, Dhaka, Bangladesh

## Abstract

Lemmatization holds significance in both natural language processing (NLP) and linguistics, as it effectively decreases data density and aids in comprehending contextual meaning. However, due to the highly inflected nature and morphological richness, lemmatization in Bangla text poses a complex challenge. In this study, we propose linguistic rules for lemmatization and utilize a dictionary along with the rules to design a lemmatizer specifically for Bangla. Our system aims to lemmatize words based on their parts of speech class within a given sentence. Unlike previous rule-based approaches, we analyzed the suffix marker occurrence according to the morpho-syntactic values and then utilized sequences of suffix markers instead of entire suffixes. To develop our rules, we analyze a large corpus of Bangla text from various domains, sources, and time periods to observe the word formation of inflected words. The lemmatizer achieves an accuracy of 96.36% when tested against a manually annotated test dataset by trained linguists and demonstrates competitive performance on three previously published Bangla lemmatization datasets. We are making the code and datasets publicly available at https://github.com/eblict-gigatech/BanLemma[1] in order to contribute to the further advancement of Bangla NLP.

## 1 Introduction

Lemmatization is a crucial task in Natural Language Processing (NLP), where the goal is to obtain the base form of a word, known as the lemma. It has widespread applications in several NLP tasks, such as information retrieval (Balakrishnan and Lloyd-Yemoh, 2014), text classification (Toman et al., 2006), machine translation (Carpuat, 2013), etc. Lemmatization is a particularly challenging task in highly inflectional languages such as Bangla (Bhattacharya et al., 2005), due to the large number of inflectional and derivational suffixes that can be added to words. Generally, lemmatization reduces the inflectional form of a word to its dictionary form.

Lemmatization in Bangla has several challenges due to various linguistic factors. Firstly, the language exhibits a wide range of morphological diversity, making it difficult for a system to cover all its aspects (Islam et al., 2022). Secondly, Bangla has approximately 50000 roots, each capable of generating a large number of inflected words based on several factors such as tense, gender, and number (Chakrabarty et al., 2016). Thirdly, Bangla words can have multiple meanings, known as polysemy, depending on their Part-of-Speech (PoS), surrounding words, context, and other factors (Chakrabarty and Garain, 2016). Additionally, the development of lemmatization systems for this complex morphological language is hindered by the lack of available resources.

Prior research attempts have employed different methodologies (e.g., learning-based, rule-based, and hybrid approaches) (Pal et al., 2015; Das et al., 2017; Islam et al., 2022; Chakrabarty and Garain, 2016). Despite the fact that some studies have shown satisfactory performance in specific scenarios, there remains a pressing need for a robust lemmatizer tailored to the Bangla language.

In this study, we have taken a rule and dictionary-based approach to tackle the lemmatization problem in Bangla. Unlike other rule-based studies for lemmatization in Bangla, we have derived the stripping methods based on the suffix marker sequences considering the PoS class of a word. A suffix দেরগুলোতে (ḍergulote) from a word শিশুদেরগুলোতে (ʃiʃuḍergulote) is a combination of দের (ḍer), গুলো (gulo), and তে (ṭe) markers. We strip the last to the first marker sequentially to ob-

---
*Corresponding author (faisal.cse06@gigatechltd.com)
[1]The repository contains the codes, analysis dataset, list of markers, test datasets, and a sample dictionary.

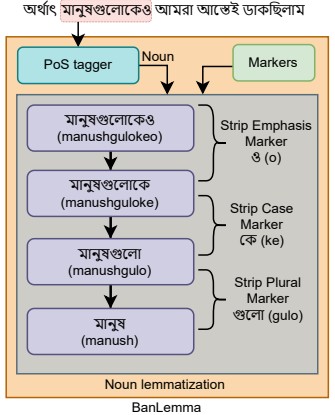

(a) Illustration of BanLemma lemmatizer

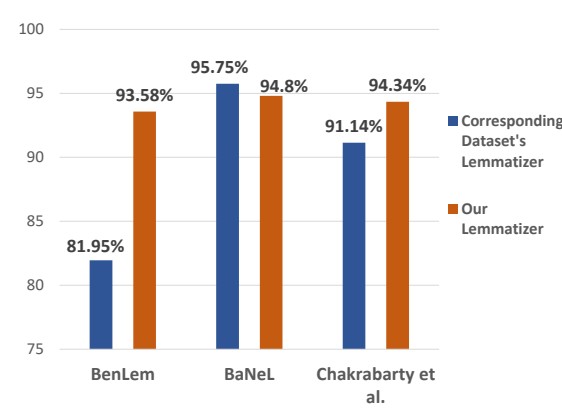

(b) Cross dataset evaluation

Figure 1: **Figure 1a** shows the illustration of our proposed BanLemma lemmatizer. For simplicity, we demonstrate Noun lemmatization only. For noun, emphasis, case, and plural markers are stripped in succession as specified in **Section 3.1**. **Figure 1a** shows its effectiveness on cross-dataset settings. On Chakrabarty and Garain (2016), we outperform their lemmatizer by a significant margin and achieve competitive results on the whole dataset of Islam et al. (2022). On Chakrabarty et al. (2017), we outperform their model on the corrected PoS and Lemma version of their evaluation set, elaborated in **Section 5.3**.

tain the lemma শিশু (ʃiʃu; child). To derive the marker sequences, we analyzed the word formations using a large Bangla text corpus. By embracing the marker sequence striping approach, we are able to effectively address a wide range of suffixes in the Bangla language, where other studies exhibit substantial limitations. **Figure 1** shows an illustration of our lemmatization process and its effectiveness in cross-dataset settings where our lemmatizer achieves higher accuracy than other lemmatizers when tested on their respective datasets (Chakrabarty and Garain, 2016; Islam et al., 2022; Chakrabarty et al., 2017). The key contributions of this study are as follows:

- We introduce BanLemma, a lemmatization system specifically designed for Bangla. By leveraging a precisely crafted linguistical framework, our system demonstrates superior performance compared to existing state-of-the-art Bangla lemmatization methods.

- We present a set of linguistical rules interpreting the process by which inflected words in the Bangla language are derived from their respective base words or lemmas.

- The linguistic rules are derived from rigorous analysis conducted on an extensive Bangla text corpus of $90.65M$ unique sentences. It encompasses a vast collection of $0.5B$ words, where $6.17M$ words are distinct. We sampled

22675 words through a systematic approach to manually analyze the inflected words.

- To assess the efficacy of BanLemma, we have employed both intra-dataset and cross-dataset evaluation. This evaluation framework enables us to measure the robustness of our proposed system across multiple datasets.

- Utilizing human annotated PoS tag, we have achieved 96.36% accuracy on intra-dataset testing. Moreover, in cross-dataset testing, BanLemma surpasses recently published methodologies, exhibiting substantial performance improvements ranging from 1% to 11% (see **Figure 1b**), which implies our proposed BanLemma's robustness.

## 2 Related Work

Preliminary works on lemmatization mainly consisted of rule-based and statistical approaches. Pal et al. (2015) created a Bangla lemmatizer for nouns where they removed non-inflected nouns using the Bangla Academy non-inflected word list (Choudhury, 2008) and removed the suffixes via the longest match suffix stripping algorithm. Das et al. (2017) created a lemmatizer for Bangla verbs according to tense and person using Paninian grammar described in Ashtadhyayi[2]. Kowsher et al.

---

[2]https://ashtadhyayi.com/

| Sentence | Word | PoS | Lemma |
|---|---|---|---|
| নিয়মিত কর দিন। (niʲomit̪o kɔr d̪ao; Pay your taxes regularly.) | কর (kɔr; taxes) | Noun | কর (kɔr; tax) |
| যা বলছি তা কর। (ɟa bolecʰi t̪a koro; Do as I say.) | কর (kɔro; do) | Verb | করা (kɔra; to do) |

Table 1: Meaning difference of a word based on its PoS class. When used as a noun, the word "কর" means "tax", while used as a verb, it means "to do".

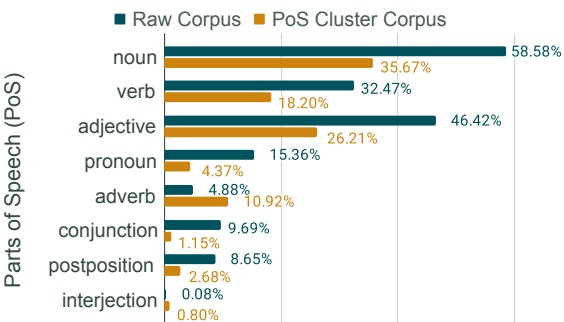

Figure 2: Distribution of all words in the raw text corpus and the analysis dataset by their PoS classes, excluding punctuation and symbols. Both corpora have a similar distribution with an abundance of nouns, verbs, and adjectives and a small number of interjections.

(2019) used two novel techniques jointly: Dictionary Based Search by Removing Affix (DB-SRA) and Trie (Cormen et al., 2009) to lemmatize Bangla words. Chakrabarty and Garain (2016) proposed a novel Bangla lemmatization algorithm using word-specific contextual information like part of speech and word sense.

Contextual lemmatizers lemmatize a word based on the surrounding context using deep neural networks. Chakrabarty et al. (2017) employed a two-stage bidirectional gated recurrent neural network to predict lemmas without using additional features. Release of the Universal Dependencies (UD) dataset (de Marneffe et al., 2014,Nivre et al., 2017) and Sigmorphon 2019 shared task formed the basis of encoder-decoder architectures to solve the lemmatization task as a string-transduction task (Qi et al., 2018; Kanerva et al., 2018). For the Bangla language, Saunack et al. (2021) employed a similar two-step attention network that took morphological tags and inflected words as input. Islam et al. (2022) used PoS tags of each word as additional features to the encoder-decoder network achieving 95.75% accuracy on validation dataset.

Earlier rule-based approaches did not consider the composition of suffixes and removed them based on the highest length or trie-like data structures. In contrast, we provide specific rules on how a sequence of markers forms a suffix based on a word's PoS tag and show its efficacy on a cross-dataset setup.

## 3 Methodology

In Bangla, the meaning of a word is greatly influenced by its PoS class within a given context of a sentence (see **Table 1**). Inflections are morphemes that convey grammatical features without changing the word class or semantic meaning (Lieber, 2021). They do not involve adding prefixes and altering the word's meaning. According to Karwatowski and Pietron (2022), for lemmatization it is crucial to determine the word's intended PoS accurately and its meaning within a sentence, considering the broader context. In this study, we have adopted a word formation-dependent rule-based approach, considering the following factors: i) The lemmatizer will operate on the inflected forms only and leave the derivational forms as they are. ii) The rules depend on the words' PoS class to use the contextual information.

### 3.1 Development of Lemmatization Rules

To analyze the behavior of inflected words, we utilized a raw text corpus of 90.65 million sentences, totaling about 0.58 billion words, where approximately 6.17 million words are unique. The corpus was crawled from 112 sources, covering ten different domains across various time periods. **Figure 3** and **Figure 4** in **Appendix A.1** provide visual representations of the dataset's distribution across domains and time respectively.

To obtain the PoS tags for each word in the dataset, we employed the automatic PoS tagger from the BNLP toolkit (Sarker, 2021). We projected each narrow PoS class to its corresponding basic PoS class: noun, pronoun, adjective, verb, adverb, conjunction, interjection, and postposition (Islam et al., 2014). For example, NC (common noun), NP (proper noun), and NV (verbal noun) were mapped to the class "noun". This allowed us to categorize the words based on their PoS classes.

After categorizing the words, we selected 19591 words as the analysis dataset which was used to

analyze the inflection patterns of Bangla words. The analysis dataset preparation procedure is elaborated in **Appendix A.2**. Word distribution of the raw text corpus and analysis dataset per PoS class is depicted in **Figure 2**. As the majority of the words in the analysis dataset were in colloquial form, there was insufficient data available to study words in classical forms, such as তাহাদিগর (ṭahaḍigɔr; their), গিয়াছিলেন (giʲacʰilen; went), etc. To address this limitation, we collected classical texts from specifically selected sources[3]. Utilizing 500 sentences comprising 5155 total words, where 3084 words were unique, we manually created clusters for the corresponding PoS classes using these classical texts. We did not employ the automatic PoS tagger here as it was trained only on colloquial text (Bali et al., 2010). Adding classical texts allowed us to include a wider range of words, with a total of 22675 words.

The morphological synthesis of different PoS is highly effective in determining whether inflections are applied and helps identify the lemma. The investigations of the analysis dataset revealed interesting sequential patterns of inflected words from different PoS classes. Nouns and pronouns were found to have four inflectional suffixes, including case markers (Moravcsik, 2008), plural markers, determiners, and emphasis markers. Verb inflections, on the other hand, depend on factors like tense, person, and number. Adjectives, in comparison, have only two suffixes তর (ṭɔro) and তম (ṭɔmo) which indicate the comparative and superlative degrees respectively (Das et al., 2020). Lastly, only emphatic inflections are found in adverb and postposition word classes. It should be noted that while other PoS classes exhibit distinct patterns of inflection, conjunctions, and interjections function without undergoing any inflection.

### 3.1.1 Inflections of Nouns

Nouns in Bangla comprise both NP and NC, which contribute a significant portion to the vocabulary of Bengali phrases. Nominal inflections are observed at four levels of nouns, including inanimate, animate, human, and elite (Faridee and Tyers, 2009). These inflections are added to nouns to signify grammatical roles and incorporate morphological features.

---

[3]Used the following sources to extract classical text:
https://bankim-rachanabali.nltr.org,
https://kobita.banglakosh.com,
https://rabindra-rachanabali.nltr.org

Case markers in the Bangla suffix system determine the noun's role in the sentence, indicating subject, object, possessor, or locative position. From seven Bangla cases, four case markers are used as noun suffixes e.g., nominative, objective, genitive, and locative (Mahmud et al., 2014). Determiner markers in Bangla noun suffixes provide specificity and indicate singularity, while plural markers indicate multiple entities or instances of a noun phrase. Some plural markers are specifically used with animate nouns, such as গণ (gɔn), বৃন্দ (brinḍo), মণ্ডলী (mɔndoli), কুল (kul), etc. while others are used with inanimate objects, like আবলি (abli), গুচ্ছ (guccʰo), গ্রাম (gram), চয় (cɔʲ), etc. (Islam and Sarkar, 2017). Though these suffixes are found in traditional grammar and literature, their frequency of usage is quite low. Emphasis markers ই (i) and ও (o) are employed to emphasize nouns. **Table 9** in **Appendix A.3** lists the markers used in nouns .

In Bengali, the word মানুষগুলোকেও (manuʃgulokeo͡) is formed by combining the base word মানুষ (manuʃ; human) + গুলো (gulo) + কে (ke) + ও (o) where lemma is মানুষ (manuʃ; human) with গুলো (gulo) plural, কে (ke) case, and ও (o) emphatic markers. This inflected form expresses the meaning of "even the humans" and conveys plurality, objective case, and emphasis in a single word. The order of these suffixes is crucial because altering the sequence, such as মানুষওকেগুলো (manuʃokegulo), would result in a nonsensical word. The key considerations in noun morphology are selecting the appropriate suffixes, figuring out how to arrange them, and understanding the changes that take place at the border during affixation (Bhattacharya et al., 2005).

From analysis, we find that the emphasis marker always takes the end position of a noun and does not occur in the middle of the suffix sequence, where the other markers can be combined in different ways. **Table 10** in **Appendix A.3** lists some examples of how nouns are inflected by taking different markers in a specific sequence in the marker combinations, as represented by **Equation 1**.

$$W = L + (PM + CM) \, || \, (CM + PM) \\ + DM + CM + EM \quad (1)$$

Here, $W$, $L$, $PM$, $CM$, $DM$, and $EM$ denote the original word, the corresponding lemma, the plural marker, case marker, determiner marker, and emphasis marker, respectively. We would use these

notations in the following equations also. According to the equation, an inflected form of a word can consist of a lemma and up to four possible suffixes. The first suffix can be either a plural marker or a case marker, and they can alternate positions. However, it is not possible for one type of marker to occur twice consecutively. It is also possible to omit any of these suffixes from the sequence.

### 3.1.2 Inflections of Verbs

Bangla extensively utilizes verbs, which are action words comprising a verb root and an inflectional ending. The inflectional ending varies based on the tense (present, past, future), person (first, second, third), and honor (intimate, familiar, formal) (Mahmud et al., 2014). Traditional Bangla grammar divides the tense categories into ten different forms. Verbs in Bangla can be written in classical/literary form or colloquial form. **Table 11** of **Appendix A.4** showcases the different forms of verbal inflection in Bangla.

In Bangla, verb suffixes do not break down into markers like other parts of speech. Removing the suffixes from a verb does not yield the lemma but rather the root form of the verb. For instance, যাচ্ছি (ɟaccʰi; going), যাবো (ɟabo; will go), and গিয়েছিলাম (giɟecʰilam; went) are different forms of the verb যাওয়া (ɟao͡ʷa; to go), which is the dictionary word (Das et al., 2020; Dash, 2000).

After stripping the suffixes from the inflected verbs যাচ্ছি (ɟaccʰi; going) and যাবো (ɟabo; will go) we would get the suffixes চ্ছি (যা + চ্ছি) (ccʰi(ɟa + ccʰi)) and বো (যা + বো) (bo(ɟa + bo)) respectively. After stripping the suffixes, we get the root যা (ɟa; go). This lemmatizer will map the root যা (ɟa; go) to the lemma যাওয়া (ɟao͡ʷa; to go). However, in Bangla verbs, some exceptions are found such as গিয়েছিলাম (giɟecʰilam; went), এসেছি (eʃecʰi), etc. After stripping the verbal inflections we get the suffixes য়েছিলাম (গি + য়েছিলাম) (ɟecʰilam(gi + ɟecʰilam)) and এসেছি (এস + এছি) (eʃecʰi (eʃ + ecʰi )) and the root গি (gi) and এস (eʃ) which does not match with the actual verb roots যা (ɟa) and আস (aʃ). In these cases, the lemmatizer will directly map the verb to the lemma using a root-form to verb-lemma mapping which is shown in **Table 2**.

Briefly, a two-step process is followed to accurately lemmatize verbs. Firstly, the suffixes are removed from the verb to extract its root form. Then, a root-to-lemma mapping is applied to determine the final lemma form of the verb.

| Word | Suffix | Root | Lemma |
|---|---|---|---|
| যাচ্ছি (ɟaccʰi) | চ্ছি (ccʰi) | যা (ɟa) | যাওয়া (ɟao͡ʷa) |
| যাবো (ɟabo) | বো (bo) | যা (ɟa) | যাওয়া (ɟao͡ʷa) |
| গিয়েছিলাম (giɟecʰilam) | য়েছিলাম (ɟecʰilam) | গি (gi) | যাওয়া (ɟao͡ʷa) |

Table 2: The two-pass approach to lemmatize verbs in Bengali. Firstly, a suffix such as ছিলাম (cʰilam; was) is removed from a word, for instance খেলছিলাম (kʰelcʰilam; was playing), to retrieve its root form খেল (kʰæl; to play). Then, the root is matched to a lemma in a root-lemma mapping to obtain the lemma form, such as খেলা (kʰela; play).

### 3.1.3 Inflections of Pronouns

Bangla pronouns represent specific nouns and exhibit similar inflectional patterns to noun classes. The language has nine types of pronouns, categorized into first, second, and third person based on personal distinctions (Dash, 2000). **Appendix A.5** lists the singular, plural, and possessive forms of Bangla personal pronouns, offering a comprehensive understanding of their usage in the language.

Many Bangla personal pronouns have inherent suffixes that are integral to the words, and stripping these suffixes can result in meaningless strings. For example, pronouns like আমার (amar; my), তোমার (ʈomar; your), আমাদের (amaḍer; our), তোমাদের (ʈomaḍer; yours) contain case markers র (rɔ) and দের (ḍer) as inherent parts, which can further be inflected with other markers like the plural marker, determiner, and emphasis marker. Additionally, other pronouns are inflected with four nominal suffixes, including the plural marker, case marker, determiner, and emphasis markers. For instance, the pronoun তোমাদেরকেই (ʈomaḍerke͡i) is inflected with the case marker কে (ke) and the emphasis marker ই (i). However, the marker দের (ḍer) is considered part of the pronoun itself, and our lemmatizer does not strip it, resulting in the lemma being তোমাদের (ʈomaḍer; yours), even though দের (ḍer) can function as a case marker. Pronoun lemmas can be inflected using the marker sequence shown in **Equation 2**.

$$W = L + PM + DM + CM + EM \quad (2)$$

### 3.1.4 Inflections of Adjectives, Adverbs and Postpositions

Bangla adjectives serve as modifiers for nouns expressing their features and can also modify adverbs. Suffix markers associated with adjectives indicate comparative and superlative degrees (Das et al., 2020). There are only two degree markers, তর (ʈɔro) as comparative and তম (ʈɔmo) as

superlative, which inflect adjectives to indicate a degree. The lemmatizer strips the degree marker from an adjective and results in the corresponding positive adjective. For example: বৃহত্তর (brihoṭṭor; largest) is lemmatized as বৃহৎ (brihoṭ; large) and ক্ষুদ্রতম (kʰuḍroṭɔmo; smallest) is lemmatized as ক্ষুদ্র (kʰuḍro; small). Adjectives can also take the form of numerics when quantifying nouns. For example, একটি (ekti; a) is an adjective inflected with a nominal suffix. In such cases, the lemmatizer will not strip the suffix, resulting in the lemma being the same as the inflected form. Moreover, because of syntactic structure, nouns can function as adjectives e.g আগের দিনের স্মৃতিগুলো (ager ḍiner sriṭigulo; The memories of the previous day). Here, আগের (ager; previous) will be unchanged as the lemma for being an adjective in this sentence. Additionally, adjectives can be inflected with emphatic markers. **Equation 3**, where $DgM$ represents the degree marker, indicates the sequence of markers that can inflect an adjective.

$$W = L + DgM + EM \qquad (3)$$

Adverbs modify verbs to indicate how an action takes place. Postpositions, on the other hand, serve a functional role in establishing syntactic connections between syntactic units (Bagchi, 2007). Postpositions can also undergo inflection by emphasis markers only. Adverbs and postpositions are inflected according to $W = L + EM$ sequence.

Our analysis revealed that words belonging to conjunction and interjection PoS classes do not get inflected in the Bangla language.

## 3.2 BanLemma

BanLemma consists of two main components: PoS-dependent rules and a dictionary. When given an input sentence, BanLemma employs an automatic PoS tagger, a suffix list, and a dictionary. The PoS tagger assigns tags to each word in the sentence, resulting in a list of word-PoS tag pairs. Subsequently, BanLemma iterates over each element of the list and applies the relevant lemmatization rule based on the PoS tag. In the case of a noun, BanLemma utilizes a method based on **Equation 1** to determine the lemma. In contrast, the method utilizes the dictionary and applies sequential suffix stripping to determine the lemma as described in **Algorithm** 1. We discuss more detailed and implementation-oriented pseudo codes in the **Appendix A.6**.

---

**Algorithm 1** Marker stripping method

---

**Require:** A word ($W$), Marker list ($M$), and Dictionary words ($D_w$)
**Ensure:** The marker list is sorted according to length in descending order.
  **function** strip_marker($W, M, D_w$)
      $L \leftarrow \text{len}(W)$
      $L_{max} \leftarrow 0$
      $m_{max} \leftarrow string()$
      **for all** $m \in M$ **do**
         **if** $W$ endswith $m$ **then**
            $w \leftarrow W[0 \ldots L - \text{len}(m)]$
            **if** $w \in D_w$ **then**
               **return** $D_w[w]$
            **else if** $\text{len}(m) > L_{max}$ **then**
               $L_{max} \leftarrow \text{len}(m)$
               $m_{max} \leftarrow m$
            **end if**
         **end if**
      **end for**
      **if** $L_{max} > 0$ **then**
         $W \leftarrow W[0 \ldots L - L_{max}]$
      **end if**
      **return** $W$
  **end function**

---

### 3.2.1 Development of BanLemma Dictionary

The dictionary used in the lemmatization process includes inflected words and their corresponding lemmas. For instance, (অংশীদারকে (ɔŋʃiḍarke; to the partner), অংশীদার (ɔŋʃiḍar; partner)) represents the mapping of an inflected word to its lemma. However, for base words, the key and value in the mapping are the same, as in (কেতন (kæṭon; flag), কেতন (kæṭon; flag)). The dictionary is organized into 6 PoS clusters (e.g., nouns, pronouns, verbs, adjectives, adverbs, and postpositions) containing a total of around 71.5k word-lemma pairs. To prepare the dictionary, we used sources including Accessible (2023); Chowdhury (2012). The dictionary format and organization are shown in **Figure 6** in **Appendix A.7**.

## 4 Experimental Setup

We evaluate BanLemma's performance using different PoS taggers: human-annotated tags, BNLP toolkit (Sarker, 2021), and ISI[4] using the Stanford Postagger[5] implementation. Additionally, we

---

[4] www.isical.ac.in/~utpal/resources.php
[5] nlp.stanford.edu/software/tagger.shtml

conduct cross-dataset evaluation and compare our methodology with existing Bangla lemmatizers, including BenLem (Chakrabarty and Garain, 2016), BaNeL (Islam et al., 2022), and Chakrabarty et al. (2017).

## 4.1 Test Dataset Preparation

We created a test dataset using the text corpus described in **Section 3.1**. Instead of random selection, we employed a systematic approach to choose 1049 sentences while maintaining the same domain distribution. The detailed procedure for preparing the test dataset is discussed in **Appendix A.8**. This dataset had 25.16% words overlapping with the analysis dataset, enabling a reliable evaluation of our proposed rules. To ensure accuracy, we manually annotated the PoS tags and lemmas of the test dataset. We also prepare a separate test dataset containing only classical texts that contain 70 sentences totaling 607 words.

## 5 Results & Analysis

### 5.1 Performance of BanLemma

**Table 3** summarizes the lemmatizer's performance for PoS categories. At first, we evaluate it using the whole test dataset and report the result in *All* column, where we achieve 96.36% overall accuracy. To measure the performance on the classical texts only, we separate the classical sentences and report the performance on the *CSCL* column that demonstrates 96.48% overall accuracy. After that, we tried to evaluate the performance of words we did not include during the manual analysis of inflected words, i.e., non-overlapping with the analysis dataset. Column *NOAD* shows we achieved an accuracy of 96.41% in this attempt. Finally, we measured the performance of the words where neither the word nor the lemma was not included in the dictionary. We report the accuracy to be 96.32% for this experiment in the *NOD* column. The table shows that the lemmatizer achieves a perfect accuracy of 100% for postpositions, which can be attributed to the limited number of lemmas and inflections in this category, most of which are included in the word-lemma mapping dictionary. The "others" category also achieves 100% accuracy, as the lemmatization process considers the word itself as the lemma for any category not explicitly targeted for lemmatization. We further evaluated the performance of BanLemma in terms of precision, recall, and F1 score, which is dis-

| PoS | Accuracy (%) | | | |
|---|---|---|---|---|
| | **All** | **CSCL** | **NOAD** | **NOD** |
| Noun | 95.20 | 94.89 | 94.60 | 90.79 |
| Pronoun | 94.28 | 93.59 | 94.12 | 87.50 |
| Verb | 95.12 | 96.58 | 84.11 | 78.26 |
| Adverb | 96.88 | 96.15 | 96.67 | 98.28 |
| Adjective | 96.93 | 98.11 | 98.40 | 97.47 |
| Postposition | 100.00 | 100.00 | - | - |
| Others | 100.00 | 100.00 | 100.00 | 100.00 |
| **Overall** | **96.36** | **96.48** | **96.41** | **96.32** |

Table 3: PoS wise and overall performance of the lemmatizer using the human-annotated test dataset. *All*, *CSCL*, *NOAD*, and *NOD* represents the dataset of all text, classical text, **N**o **O**verlap with **A**nalysis **D**ataset and **N**o **O**verlap with **D**ictionary respectively. *Others* indicates all other PoS tags except the aforementioned PoS tags. For the "others" class, the word itself is considered as its lemma and there was no non-overlapping Postposition for NOAD and NOD datasets.

cussed in **Appendix A.9**.

### 5.2 Dependency on Automatic PoS Tagger

| Metric | Human annotated PoS | BNLP PoS tagger | ISI PoS tagger |
|---|---|---|---|
| Accuracy (%) | 96.67 | 89.32 | 84.77 |

Table 4: Lemmatizer's performance based on how PoS tags were obtained.

**Table 4** summarizes the impact of using an automatic PoS tagger to get the tags of each word. It indicates that the lemmatizer tends to show significantly reduced performance when used with an automatic PoS tagger. It also highlights that the rules are well capable of lemmatizing Bangla text more accurately given the correct PoS. **Table 5** provides examples where the lemmatizer fails to generate accurate lemmas due to incorrect PoS information. These examples highlight the dependency of the lemmatizer on the accuracy of the automatic PoS tagger. It demonstrates that the lemmatizer is capable of producing the correct lemma when provided with accurate manually annotated PoS tags.

### 5.3 Cross Dataset Evaluation

**Table 6** presents the results of the cross-dataset evaluation. Our lemmatizer outperforms BenLem, achieving an 11.63% improvement in accuracy on their provided test dataset. BaNeL did not provide any separate test dataset. So, we evalu-

| Word | Target lemma | Automatically Predicted PoS tag | Lemma with predicted PoS tag | Manually Annotated PoS tag | Lemma with annotated PoS tag |
|---|---|---|---|---|---|
| সবাই (ʃɔbaĩ) | সবাই (ʃɔbaĩ) | adjective | সবা (ʃɔba) | pronoun | সবাই (ʃɔbaĩ) |
| ভালবাসি (bʰalbaʃi) | ভালবাসা (bʰalbaʃa) | adjective | ভালবাসি (bʰalbaʃi) | verb | ভালবাসা (bʰalbaʃa) |
| হাসান (hasan) | হাসান (hasan) | verb | হাসা (haʃa) | noun | হাসান (hasan) |

Table 5: The table showcases instances where our lemmatizer produces incorrect lemmas when an automatic PoS tagger provides inaccurate tags but accurately lemmatizes a word if the PoS tag is correct.

ated the lemmatizer on the entire BaNeL dataset. Though they reported the performance on a test split, our lemmatizer demonstrates competitive performance achieving 94.80% accuracy on the whole dataset, which is only 0.95% less than their reported accuracy. On the other hand, our system exhibits lower performance on the test dataset provided by Chakrabarty et al. (2017), achieving an accuracy of 80.08%, which is 11.06% lower than the reported performance. Several factors contribute to this performance gap, including the reliance on an automatic PoS tagger, which introduced inherent errors. Further investigation reveals significant inconsistency between the dataset used in their study and our considerations during the development of the lemmatizer. These inconsistencies are discussed in detail in **Appendix A.10**.

| Test dataset | Study | Acc | Ch Acc |
|---|---|---|---|
| BenLem | BenLem | 81.95 | - |
| | Ours | 93.58 | |
| BaNeL | BaNeL | - | 95.75 |
| | Ours* | - | 94.80 |
| Chakrabarty et al. | Chakrabarty et al. | 91.14 | - |
| | Ours | 80.08 | |

Table 6: Lemmatization results on cross-dataset evaluation. * We report the performance on the entire dataset of BaNeL while they reported the metric on a test split from the entire dataset (*Acc*=Accuracy and *Ch Acc*=Character Accuracy).

To evaluate the performance of our proposed system on the dataset provided by Chakrabarty et al. (2017), we manually annotated and corrected PoS tags and lemmas. We focused on 52 sentences comprising a total of 695 words. Additionally, we reviewed and corrected 2000 lemmas from the BaNeL dataset. The results of our evaluation are summarized in **Table 7**.

For the Chakrabarty et al. (2017) dataset, we

| Dataset | Acc. | A-PoS | C-PoS C-Lem. |
|---|---|---|---|
| Chakrabarty et al. | 79.97 | 87.09 | 94.34 |
| BaNeL | 96.36 | - | 98.99 |

Table 7: Performance of our lemmatizer on the sampled cross dataset measured in accuracy. The second column, *Acc.* indicates accuracy on the unmodified datasets. *C-PoS* and *A-PoS* indicates manually and automatically annotated PoS tags. *C-Lem.* indicates manually corrected lemmas.

performed three evaluation steps. Firstly, we assessed the performance of our lemmatizer on unmodified data within the portion where manual efforts were applied. The accuracy column (*Acc.*) of the table presents the accuracy on this attempt to be 79.97%. Secondly, we examined the performance using an automatic PoS tagger while correcting the lemmas. The automatic PoS tags (*A-PoS*) and original lemma (*O-Lem*) column report this accuracy to be 87.09%. Finally, we measured the overall performance using manually annotated PoS tags and corrected lemmas. The correct PoS (*C-PoS*) and corrected lemma (*C-Lem*) column of the table illustrate the outcomes of the final experiment to be 94.34%.

Since the BaNeL dataset already provides manually annotated PoS tags, we focused solely on evaluating the performance of our lemmatizer on the corrected lemmas. The fourth column of the table presents the performance in this scenario. In both cases, our lemmatizer demonstrated improved performance compared to the initial evaluations.

## 6 Conclusion and Future Works

This study introduces BanLemma, a Bangla lemmatization system aimed at enriching Bangla language resources. BanLemma is composed of linguistically derived rules, obtained through rigorous analysis of a large Bangla text corpus. To

overcome the challenges associated with limited resources in Bangla lemmatization, we also provide a comprehensive collection of morphological markers and rules. To demonstrate the effectiveness of BanLemma, we have conducted evaluations using a human-annotated test dataset, annotated by trained linguists and some recently published Bangla datasets. Our proposed BanLemma achieved an accuracy of 96.36% on our human-annotated test set. Furthermore, in cross-dataset evaluation, BanLemma exhibited significant performance improvements ranging from 1% to 11%. The results of our study shed light on the formation of inflected words, offering a solution to address the limitations of previous lemmatization methods. These findings contribute to the advancement of research in this field and pave the way for further investigations in the domain of Bangla lemmatization.

## Limitations

During our analysis, we found some limitations of BanLemma. We discuss these in the following points:

- **Out of dictionary words:** we identified a notable pattern in the lemmatizer's behavior regarding words that are not present in the dictionary but already are lemmas and end with a suffix substring. In this case, the lemmatizer erroneously strips the suffix from the words. For instance, the word নূন্যতম (nunnɔṭɔmɔ; minimum) itself is a lemma, yet the lemmatizer strips the ending substring তম (ṭɔmo), resulting in the lemma নূন্য (nunno), which is incorrect. We also noticed that this limitation is particularly prominent with proper nouns. It also emphasizes the significance of the dictionary's richness in the lemmatization process. Words that are not present in the dictionary but end with suffix substrings will be inaccurately lemmatized.

- **Ambiguous semantic meaning:** We observed that the lemmatizer struggles to comprehend the semantic meaning of certain words, resulting in incorrect lemmatization. For example, **Table 8** illustrates a case where the lemma varies depending on the context although having the same PoS class. The lemmas differ based on whether they express the

action of hanging or the state of something being hung.

- **Automatic PoS dependency:** The lemmatizer heavily relies on PoS information, which introduces errors when an automatic PoS tagger is used in the workflow.

| Sentence | Word | Lemma |
|---|---|---|
| দড়িটি ঝুলিয়ে দাও। (doṛiti ɟʰuliʲe da̯o; Hang the rope) | ঝুলিয়ে (ɟʰuliʲe) | ঝুলানো (ɟʰulano) |
| দড়িটি ঝুলছে। (doṛiti ɟʰulcʰe; The rope is hanging) | ঝুলছে (ɟʰulcʰe) | ঝুলা (ɟʰula) |

Table 8: Difference of lemma of the same verb ঝুলা (ɟʰula) based on the context of the sentences. The lemma is ঝুলানো (ɟʰulano) when it expresses an action of hanging something, and the lemma is ঝুলা (ɟʰula) when it expresses that something is hanging.

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

## A  Appendix

### A.1  Raw Corpus Distribution

**Figure 3** and **Figure 4** provide representations of the raw corpus's distribution across domains and time respectively.

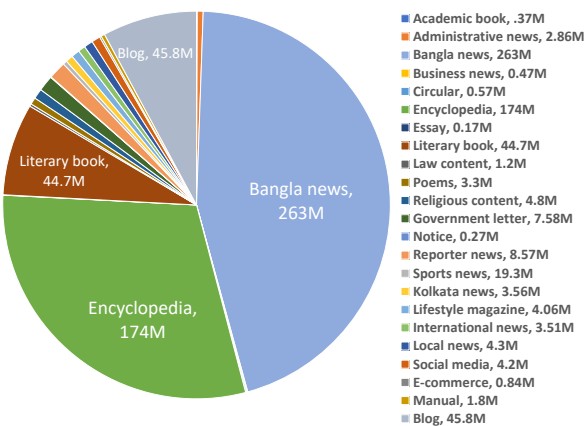

Figure 3: Distribution of word count of raw dataset corpus categorized by their respective domain. The amount of word count is denoted by 'M' (millions). The majority of the data originates from the "Bangla News" and "Encyclopedia" domains, while the "Essay" data represents the smallest portion. However, there are 66757 words, accounting for 0.01% of all words, for which the domain could not be determined.

### A.2  Analysis Dataset Preparation

To obtain the PoS tags for each word in the dataset, we used the PoS tagger from the BNLP toolkit. The tagger was trained on the *Indian Language Part-of-Speech Tagset: Bengali LDC2010T16* (Bali et al., 2010) dataset, which comprises 30 narrow PoS classes. After projecting the narrow PoS classes, we grouped the words based on their PoS class for further analysis which formed the basis of our investigation into the behavior of inflected words in Bangla. In order to conduct a detailed analysis of the words, we employed a systematic approach to create a representative dataset. Initially, we clustered the words based on their longest common initial substring within each PoS group. For example, words like সরকার (ʃɔrkar; government), সরকারই (ʃɔrkari), সরকারও (ʃɔrkaro),

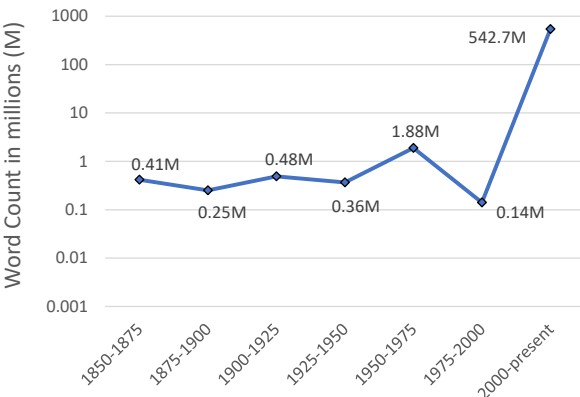

Figure 4: The time distribution of the raw text corpus is shown in the figure, where the horizontal axis is displayed on a logarithmic scale. A significant portion of the data spans from 2000 to the present, while a relatively small amount of data exists for the periods before 2000. Furthermore, approximately 7.79% of all words (46173307 words) do not have a specific time frame associated with them.

সরকারকে (ʃɔrkarke), and so on, which share the common initial substring সরকার (ʃɔrkar; government), were grouped together. For the analysis, we then selected clusters that contain a minimum threshold of words. Initially, we set a minimum threshold of 10, but this resulted in an overabundance of nouns, verbs, and adjectives while filtering out clusters from pronouns, postpositions, conjunctions, and interjections due to limited words in those groups. To address this, we individually determined the minimum threshold for each PoS class. The thresholds were set as follows: 14 for nouns, 7 for adjectives, 6 for verbs, 2 for pronouns and adverbs, and 1 for postpositions, conjunctions, and interjections. These thresholds were determined through a combination of tuning and manual examination of the selected clusters. To neutralize the error of the automatic PoS tagger, we manually curated the words while removing any word if necessary. Finally, we came up with 19591 selected words for the analysis dataset. To study the classical texts rigorously, we additionally use some classical text sources and select some words. Subsequently, we selected a total of 22675 words for further analysis.

### A.3  Markers and Noun Formation

### A.4  Verb Suffixes

**Table 11** presents all suffixes that inflect the a verb root.

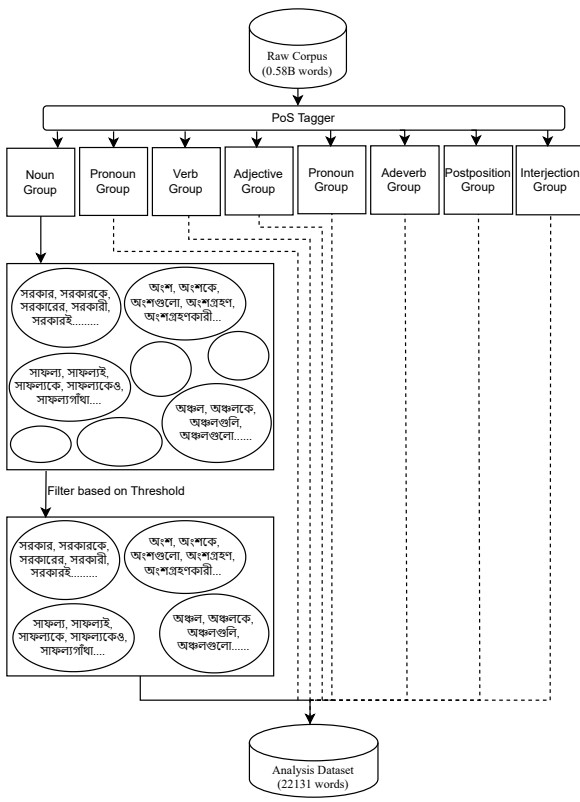

Figure 5: The analysis dataset preparation process.

| Type | Markers |
|---|---|
| Plural | আবলি (aboli), কুল (kul), গণ (gɔn), গুচ্ছ (guccʰo), গুলা (gula), গুলি (guli), গুলো (gulo), দের (ḍer), গ্রাম (gram), চয় (cɔʲ), জাল (ɟal), ত্রয় (ṭroʲ), দল (ḍɔl), দাম (ḍam), দিগ (ḍig), দিগর (ḍigɔr), দ্বয় (ḍʲ), নিকর (nikɔr), নিচয় (nicɔʲ), পাল (pal), পুঞ্জ (punɟo), বর্গ (bɔrgo), বৃন্দ (brinḍo), ব্রজ (broɟo), মণ্ডল (mɔndol), মণ্ডলী (mɔndoli), মহল (mɔhol), মালা (mala), যূথ (ɟutʰ), রা (ra), রাজি (raɟi), রাশি (raʃi), শ্রেণি (sreni), সমূহ (ʃɔmuho), সহ (ʃɔho), েরা (era), ৗচ্চয় (occɔʲ) |
| Case | কার (kar), কারে (kare), কে (ke), কের (ker), তে (ṭe), র (ro), রে (re), ে (e), েতে (eṭ), ের (er), য (ʲ), যে (ʲe) |
| Determiner | খানা (kʰana), খানি (kʰani), টা (ta), টি (ti), টুকু (tuku), টুকুন (tukun), টে (te) |
| Emphasis | ই (i), ও (o) |

Table 9: A list of markers in used in nouns which contains 37 plural markers, 12 case markers, 7 determiners, and 2 emphasis markers, totaling 58 markers.

## A.5 Personal Pronouns

**Table 12** lists the personal pronouns in singular, plural, and possessive forms.

## A.6 Lemmatization Algorithm and Methods

**Algorithm 2** summarizes the overall lemmatization algorithm.

**Algorithm 3** provides a summary of the lemmatization method for noun words. The stripping process begins with the last marker, which in the case of nouns is the emphasis marker. **Equation 1** illustrates that the sequence of the last three markers is fixed. However, if the first marker is a plural marker, then the second marker will be a case marker, or vice versa. After stripping the last three types of markers, the algorithm determines the next last marker and selects a second appropriate sequence for stripping the remaining markers.

The marker stripping method is described in detail in **Algorithm 1**. This algorithm is responsible for identifying and removing markers at the end of a word. It begins by checking if a word ends with a marker. If a match is found, it determines whether to stop the matching process based on whether the remaining prefix of the word is present in the dictionary. If the remaining prefix is found in the dictionary, the lemma is immediately returned. However, if the remaining prefix is not found in the dictionary, the algorithm continues matching to determine if stripping a shorter suffix marker would result in a dictionary entry. This design choice allows for handling cases where multiple markers are present. For example, when stripping case markers from the inflected word ছেলের (of the boy), the algorithm would first match the marker এর (ের). Stripping this marker would result in the word ছেল, which is not the correct lemma. By continuing the matching process, the algorithm would then match the marker □, resulting in the correct lemma ছেলে (boy). However, if any shorter marker is not found, it would strip the longest marker at the end.

The lemmatization methods for other PoS classes can be achieved by modifying **Algorithm 3**. The details of these modifications are discussed in **Algorithm 4** to **Algorithm 7**. **Algorithm 4** to **7** presents the algorithms to lemmatize the words from the corresponding PoS class.

| Word (noun) | Lemma | Plural | Case | Plural | Determiner | Case | Emphasis |
|---|---|---|---|---|---|---|---|
| জনগণই (ɉɔngɔni) | জনগণ (ɉɔngɔn) | | | | | | ই (i) |
| শিক্ষককে (ʃikkʰɔkke) | শিক্ষক (ʃikkʰɔk) | | | | | কে (ke) | |
| মানুষকেই (manuʃkei) | মানুষ (manuʃ) | | | | | কে (ke) | ই (i) |
| মেয়েটিকে (mejetike) | মেয়ে (mejeti) | | | | টি (ti) | কে (ke) | |
| গাছটাতেও (gacʰʈateo) | গাছ (gacʰ) | | | | টা (ta) | তে (te) | ও (o) |
| শিশুদেরটাতেও (ʃiʃuderʈateo) | শিশু (ʃiʃu) | | | দের (der) | টা (ta) | তে (te) | ও (o) |
| মায়েদেরকেও (majederkeo) | মা (ma) | | য়ে (je) | দের (der) | | কে (ke) | ও (o) |
| মায়েদেরটাতেও (majederʈateo) | মা (ma) | | য়ে (je) | দের (der) | টা (ta) | তে (te) | ও (o) |
| ভাইয়েরা (bʰaijera) | ভাই (bʰai) | | য়ে (je) | রা (ra) | | | |
| বালকগুলো (balokgulo) | বালক (balok) | গুলো (gulo) | | | | | |
| বইগুলিতেই (boigulitei) | বই (boi) | গুলি (guli) | তে (te) | | | | ই (i) |

Table 10: Examples of Bangla words which are formed with different sequences of noun suffixes to make meaningful words.

| Person & Forms | | | Present (Simple) | Present (Cont.) | Present (Compl.) | Past (Simple) | Past (Cont.) | Past (Compl.) | Past (Habitual) | Future (Simple) | Future (Cont.) VNF | Future (Compl.) VNF |
|---|---|---|---|---|---|---|---|---|---|---|---|---|
| 1st Person | | Co. | ই (i) | ছি (cʰi) | এছি (ecʰi) | লাম (lam) | ছিলাম (cʰilam) | এছিলাম (ecʰilam) | তাম (ʈam) | বো (bo) | তে (te) | এ (e) |
| | | Cl. | ইব (ib) | ইতেছি (itecʰi) | ইয়াছি (iʲacʰi) | ইলাম (ilam) | ইতেছিলাম (itecʰilam) | ইয়াছিলাম (iʲacʰilam) | ইতাম (item) | ইবো (ibo) | ইতে (ite) | ইয়া (iʲa) |
| 2nd Person | In. | Co. | কর (kor) | ছিস (cʰiʃ) | এছিস (ecʰiʃ) | লি (li) | ছিলি (cʰili) | এছিলি (ecʰilo) | তি (ʈi) | বি (bi) | তে (te) | এ (e) |
| | Fm. | Co. | ও (o) | ছো (cʰo) | এছো (ecʰo) | লে (le) | ছিলে (cʰile) | এছিলে (ecʰilo) | তা (ʈa) | বে/ বা (be/ba) | তে (te) | এ (e) |
| | | Cl. | | ইতেছ (itecʰo) | ইয়াছো (iʲacʰo) | ইলে (ile) | ইতেছিলে (itecʰilo) | ইয়াছিলে (iʲacʰilo) | ইতে (ite) | ইবে (ibe) | ইতে (ite) | ইয়া (iʲa) |
| | Fr. | Co. | এন (en) | ছেন (cʰen) | এছেন (ecʰen) | লেন (len) | ছিলেন (cʰilen) | এছিলেন (ecʰilen) | তেন (ʈen) | বেন (ben) | তে (te) | এ (e) |
| | | Cl. | | ইতেছেন (itecʰen) | ইয়াছেন (iʲacʰen) | ইলেন (ilen) | ইতেছিলেন (iʲacʰilen) | ইয়াছিলেম (iʲacʰilem) | ইতেন (iʈen) | ইবেন (iben) | ইতে (ite) | ইয়া (iʲa) |
| 3rd Person | In. | Co. | এ (e) | ছে (cʰe) | এছে (ecʰe) | লো (lo) | ছিলো (cʰilo) | এছিলো (ecʰilo) | তো (ʈo) | বে (be) | তে (te) | এ (e) |
| | | Cl. | | ইতেছেন (itecʰen) | ইয়াছে (iʲacʰe) | ইলো (ilo) | ইতেছিল (itecʰilo) | ইয়াছিল (iʲacʰilo) | ইতো (iʈo) | ইবে (ibe) | ইতে (ite) | ইয়া (iʲa) |
| | Fm. | Co. | এ (e) | ছে (cʰe) | এছে (ecʰe) | লো (lo) | ছিলো (cʰilo) | এছিলো (ecʰilo) | তো (ʈo) | বে (be) | তে (te) | এ (e) |
| | | Cl. | | ইতেছে (itecʰe) | ইয়াছে (iʲacʰe) | ইলো (ilo) | ইতেছিল (itecʰilo) | ইয়াছিল (iʲacʰilo) | ইতো (iʈo) | ইবে (ibe) | ইতে (ite) | ইয়া (iʲa) |
| | Fr. | Co. | এন (en) | এন (en) | এছেন (ecʰen) | লেন (len) | ছিলেন (cʰilen) | এছিলেন (ecʰilen) | তেন (ʈen) | বেন (ben) | তে (te) | এ (e) |
| | | Cl. | | ইতেছেন (itecʰen) | ইয়াছেন (iʲacʰen) | ইলেন (ilen) | ইতেছিলেন (itecʰilen) | ইয়াছিলেন (iyacʰilen) | ইতেন (iʈen) | ইবেন (iben) | ইতে (ite) | ইয়া (iʲa) |

Table 11: List of all suffixes that inflect the root কর (kor) of the verb করা (kora) depending on the tense, person, and honor. This covers the colloquial form (Co.) and the extended verb forms of the Bangla classical style (Cl.). The table also includes suffixes for intimate (In.), Familiar (Fm.), and Formal (Fr.) endings. In addition, the table distinguishes between the Continuative (Cont.) and Completive (Compl.) aspects of the tense.

| Person | Style | Singular | Possessive singular | Plural | Possessive plural |
|---|---|---|---|---|---|
| First | Colloquial | আমি (ami), আমাকে (amake) | আমার (amar), আমাকে (amake) | আমরা (amra) | আমাদের (amader) |
| | Classical | | আমায় (amaʲ) | | |
| Second | Colloquial | তুমি (ʈumi), তুই (ʈui), আপনি (apni) | তোমার (ʈomar), তোর (ʈor), আপনার (apnar), তোমাকে (ʈomake) | তোমরা (ʈomra), তোরা (ʈora), আপনারা (apnara) | তোমাদের (ʈomader), তোদের (ʈoder), আপনাদের (apnader) |
| | Classical | | তোমায় (ʈomaʲ) | | |
| Third | Colloquial | সে (she), তিনি (ʈini), এ (e), ও (o), উনি (uni) | তাঁর (ʈar), এর (er), ওর (or), উনার (unar) | তারা (ʈara), এরা (era), ওরা (ora) | তাদের (ʈader), এদের (eder), ওদের (oder) |
| | Classical | | তাহার (ʈahar), ইহার (ihar), উহার (uhar) | তাহারা (ʈahara), ইহারা (ihara), উহারা (uhara) | তাহাদের (ʈahader), ইহাদের (ihader), উহাদের (uhader) |

Table 12: List of personal pronouns in singular, plural, and possessive form where suffixes are included with the base form of the word.

**Algorithm 2** The lemmatization algorithm

**Require:** A sentence ($T$), PoS Tagger ($p\_tagger$), Suffix and Marker list ($S$), and Dictionary ($D$)

**Ensure:** The PoS tagger ($p\_tagger$) returns a list of tuples where the first element is the word and the second element is the PoS tag. Suffix lists are clustered into PoS classes and sorted according to length in descending order.

   **procedure** lemmatize($T, p\_tagger, S, D$)
      $lemmas \leftarrow list()$
      $words\_with\_tags \leftarrow p\_tagger(T)$
      **for all** $(W, tag) \in words\_with\_tags$ **do**
         **if** $tag = noun$ **then**
            $L \leftarrow$ noun_lemma($W, S, D$)
         **else if** $tag = pronoun$ **then**
            $L \leftarrow$ pro_lemma($W, S, D$)
         **else if** $tag = verb$ **then**
            $L \leftarrow$ verb_lemma($W, S, D$)
         **else if** $tag = adverb$ **then**
            $L \leftarrow$ adverb_lemma($W, S, D$)
         **else if** $tag = adjective$ **then**
            $L \leftarrow$ adj_lemma($W, S, D$)
         **else if** $tag = postposition$ **then**
            $L \leftarrow$ postpos_lemma($W, S, D$)
         **else**
            $L \leftarrow W$
         **end if**
         $lemmas.add(L)$
      **end for**
      $lemma\_sent \leftarrow$ space_join($lemmas$)
      **return** $lemma\_sent$
   **end procedure**

---

**Algorithm 3** Noun lemmatization method

**Require:** A noun word ($W$), Clustered marker list ($S$), and Dictionary ($D$)

**Ensure:** The word is a noun. Suffix lists are clustered into markers in a hash map where the key is the marker name and the value is a list of markers.

   **function** noun_lemma(W, S, D)
      $D_w \leftarrow D[nouns]$
      **if** $W \in D_w$ **then**
         **return** $D_w[W]$
      **end if**
      $StripSeq \leftarrow [em, cm, dm]$
      **for all** $m \in StripSeq$ **do**
         $W \leftarrow$ strip_marker($W, S[m], D_w$)
         **if** $W \in D_w$ **then**
            **return** $D_w[W]$
         **end if**
      **end for**
      $SecondStripSeq \leftarrow list()$
      **for all** $m \in S[pm]$ **do**
         **if** $W$ endswith $m$ **then**
            $SecondStripSeq \leftarrow [pm, cm]$
            **break**
         **end if**
      **end for**
      **if** len($SecondStripSeq$) $= 0$ **then**
         $SecondStripSeq \leftarrow [cm, pm]$
      **end if**
      **for all** $m \in SecondStripSeq$ **do**
         $W \leftarrow$ strip_marker($W, S[m], D_w$)
         **if** $W \in D_w$ **then**
            **return** $D_w[W]$
         **end if**
      **end for**
      **return** $W$
   **end function**

---

### A.7 The Dictionary Format

In total, the dictionary contains 6 PoS clusters such as nouns, pronouns, verbs, adverbs, adjectives, and postpositions and consist of $46, 289, 499, 5, 366, 17, 040, 860$, and $1, 353$ word-lemma pairs, respectively. The dictionary format and organization is shown in **Figure 6**

### A.8 Test Dataset Preparation

First, we divided the sentences into their respective domains and randomly reshuffled them. Then, we selected a percentage of sentences from each domain based on its contribution to the total percentage of sentences in the entire raw text corpus. For example, we sampled 452 sentences from the Bangla news domain, which accounted for 45.18% of the entire dataset. During the selection process, we made sure to include at least 1% of sentences from each domain. This decision enabled us to incorporate sentences from domains that were underrepresented in the dataset, such as circular, which accounted for only 0.09% of the entire corpus. Additionally, we included sentences for which we could not determine a specific domain. Throughout this procedure, we maintained the uniqueness of the selected sentences.

At this stage, we discovered that the test dataset had a significant overlap of 38.27% with the words used during the inflected word analysis. To ensure a more effective evaluation of our proposed

**Algorithm 4** Pronoun lemmatization method

**Require:** A pronoun word ($W$), Clustered marker list ($S$), and Dictionary ($D$)
**Ensure:** The word is a pronoun. Suffix lists are clustered into markers in a hash map where the key is the marker name and the value is a list of markers.
    **function** pro_lemma(W, S, D)
        $D_w \leftarrow D[pronouns]$
        **if** $W \in D_w$ **then**
            **return** $D_w[W]$
        **end if**
        $StripSeq \leftarrow [em, cm, dm, pm]$
        **for all** $m \in StripSeq$ **do**
            $W \leftarrow$ strip_marker($W, S[m], D_w$)
            **if** $W \in D_w$ **then**
                **return** $D_w[W]$
            **end if**
        **end for**
        **return** $W$
    **end function**

**Algorithm 5** Adjective lemmatization method

**Require:** A adjective word ($W$), Clustered marker list ($S$), and Dictionary ($D$)
**Ensure:** The word is an adjective. Suffix lists are clustered into markers in a hash map where the key is the marker name and the value is a list of markers.
    **function** ADJ_LEMMA(W, S, D)
        $D_w \leftarrow D[adjectives]$
        **if** $W \in D_w$ **then**
            **return** $D_w[W]$
        **end if**
        $StripSeq \leftarrow [em, dgm]$
        **for all** $m \in StripSeq$ **do**
            $W \leftarrow$ strip_marker($W, S[m], D_w$)
            **if** $W \in D_w$ **then**
                **return** $D_w[W]$
            **end if**
        **end for**
        **return** $W$
    **end function**

**Algorithm 6** Adverb lemmatization method

**Require:** An adverb word ($W$), Clustered marker list ($S$), and Dictionary ($D$)
**Ensure:** The word is an adverb. Suffix lists are clustered into markers in a hash map where the key is the marker name and the value is a list of markers.
    **function** ADVERB_LEMMA(W, S, D)
        $D_w \leftarrow D[adverb]$
        **if** $W \in D_w$ **then**
            **return** $D_w[W]$
        **end if**
        $StripSeq \leftarrow [em]$
        **for all** $m \in StripSeq$ **do**
            $W \leftarrow$ strip_marker($W, S[m], D_w$)
            **if** $W \in D_w$ **then**
                **return** $D_w[W]$
            **end if**
        **end for**
        **return** $W$
    **end function**

**Algorithm 7** Postposition lemmatization method

**Require:** A postposition word ($W$), Clustered marker list ($S$), and Dictionary ($D$)
**Ensure:** The word is an adverb or postposition. Suffix lists are clustered into markers in a hash map where the key is the marker name and the value is a list of markers.
    **function** POSTPOS_LEMMA(W, S, D)
        $D_w \leftarrow D[postposition]$
        **if** $W \in D_w$ **then**
            **return** $D_w[W]$
        **end if**
        $StripSeq \leftarrow [em]$
        **for all** $m \in StripSeq$ **do**
            $W \leftarrow$ strip_marker($W, S[m], D_w$)
            **if** $W \in D_w$ **then**
                **return** $D_w[W]$
            **end if**
        **end for**
        **return** $W$
    **end function**

rules, we aimed to reduce this overlap percentage. **Figure 2** illustrates that the raw corpus consists of a large number of nouns, verbs, and adjectives. Therefore, during the selection of test sentences, we excluded any sentence that contained any noun, verb, or adjective that was already in-

cluded in the analysis dataset. However, we found that the sample sentences were not well formed as the number of verbs and adjectives is limited. So, finally, we attempt to reduce the overlap by allowing all verbs and adjectives while controlling the overlapping of nouns. As a result, the final

```
{
    nouns: {
        word_1: lemma_1,
        …,
        word_N: lemma_N
    },
    verbs: {
        word_N+1: lemma_N+1,
        …,
        word_N+M: lemma_N+M
    }
    …
    PoS_P: {…}
}
```

Figure 6: The dictionary format utilized in the lemmatizer implementation. It consists of a hash map with PoS class names as keys and another hash map as values. The keys of each PoS class hash map are words and values are the corresponding word's lemma.

test dataset had only 25.16% overlapping words with the analysis dataset, where we found 9.68% nouns overlaps with the analysis dataset. However, this reduced overlapping dataset allows us to conduct a more robust evaluation. Finally, to complete the annotation process, we manually assigned PoS tags and lemmas to the words extracted from these sentences. We collaborated with an annotator who assigned the PoS tags and lemmas to each word. To ensure the accuracy and consistency of the annotations, the assigned tags and lemmas were validated by a validator who was a linguistic expert.

## A.9  Further Performance Evaluation

We were interested in evaluating how the lemmatizer works on the non-inflected and inflected words. For a non-inflected word, the word itself is the lemma. Except for the proper nouns, Usually, the non-inflected words are found in a dictionary. To conduct this experiment, we first lemmatize the sentences from the test dataset. Then separate the non-inflected and inflected words along with the lemmas. In this setup, there are a total of 6906 non-inflected words and 3125 inflected words. We found that the lemmatizer achieves an F1 score of 0.9733 for non-inflected words and 0.9399 for inflected words. **Table 13** summarizes the analysis.

| Split | Precision | Recall | F1 |
|---|---|---|---|
| Non-inflected | 0.9784 | 0.9682 | 0.9733 |
| Inflected | 0.929 | 0.9512 | 0.9399 |

Table 13: Performance of the lemmatizer on non-inflected and inflected words.

## A.10  Dataset Annotation Inconsistencies

From the dataset of BenLem, firstly we found that they labeled verb roots as lemmas, while we consider the dictionary form as the lemma. For example, they annotated the lemma of the word হবে (hɔbe) as হ (hɔ), whereas we lemmatize it as হওয়া (hɔʷa). Secondly, they converted colloquial pronouns to their classical forms. They labeled the lemma of তাদের (ṭaḍer; their) as the classical form তাহাদের (ṭahaḍer; their), whereas we consider the same colloquial form তাদের (ṭaḍer; their) as the lemma. Lastly, they made spelling changes to certain words, such as transforming ভাল (bʱalo) to ভালো (bʱalo), which differs from our approach.

During our analysis of the BaNeL dataset, we discovered the following inconsistencies. Firstly, certain derivational markers were removed. Secondly, pronoun forms were modified, converting এঁদের (eḍer; their) to তিনি (ṭini; he/she). Thirdly, spelling changes were made to some words, such as lemmatizing সেবকাধমের (ʃebokadʱɔmer) as সেবকঅধম (ʃebokodʱɔm), whereas we consider it as সেবকাধম (ʃebokadʱɔm). Additionally, incorrect lemmas were found in the dataset, where ভালোমানুষের (bʱalomanuʃer) was provided as the lemma for ভালোমানুষে (bʱalomanuʃe). Furthermore, our lemmatizer has a limitation that produces incorrect results when the actual word ends with a suffix marker. For instance, the lemma of the word জেলের (ɟeler) should be জেলে (ɟele; fisherman), but our system incorrectly lemmatizes it as জেল (ɟel; prison).

In the dataset of Chakrabarty et al. (2017) They made changes to the gender of words, such as transforming যুবতী (ɟuboṭi; young girl) to যুবক (ɟubok; young boy), altered negated forms to positive forms, e.g., changing অদূর (ɔḍur; not so far) to দূর (ḍur; far), and modified pronouns, e.g., তোমার (ṭomar; your) to তুমি (ṭumi; you). They also made derivational changes, such as transforming বাণিজ্যিক (baniɟɟik; commercial) to বাণিজ্য (baniɟɟo; trade), প্রকৃতি (prokriṭi; nature) to প্রকৃত (prokriṭo; real), and so on. These discrepancies significantly impacted the performance of our lemmatizer.