# OpenReview forum: "BanLemma: A Word Formation Dependent Rule and Dictionary Based Bangla Lemmatizer"
_EMNLP/2023/Conference — EMNLP 2023 Findings_

### Official Review · Reviewer_ATsq · 2023-08-03

**Soundness:** 2

**Excitement:**

3: Ambivalent: It has merits (e.g., it reports state-of-the-art results, the idea is nice), but there are key weaknesses (e.g., it describes incremental work), and it can significantly benefit from another round of revision. However, I won't object to accepting it if my co-reviewers champion it.

**Paper Topic And Main Contributions:**

This paper is about lemmatization for Bangla. The authors propose rule- and dictionary-based approach for lemmatization to design BanLemma lemmatization system. The authors develop new method based on the suffix marker sequences considering of the part-of-speech class of the word.

**Reasons To Accept:**

BanLemma achieved a high accuracy 96.36% compared to existing state-of-the-art Bangla lemmatization methods. The lingustic rules were derived from analysis of Bangla text corpus of 90.65M unique sentences.

**Reasons To Reject:**

Dicitionary-based approach for lemmatization has some limitations: the words out of the dictionary may incorrectly lemmatized. The authors evaluate their lemmatizer on the correct lemmas (only accuracy).

**Reproducibility:**

2: Would be hard pressed to reproduce the results. The contribution depends on data that are simply not available outside the author's institution or consortium; not enough details are provided.

**Reviewer Confidence:**

3: Pretty sure, but there's a chance I missed something. Although I have a good feel for this area in general, I did not carefully check the paper's details, e.g., the math, experimental design, or novelty.

---

> ### Author Rebuttal · Authors · 2023-08-28
>
> Thank you for your valuable feedback, and we believe these will improve the quality of the paper. We are writing our response to each point separately below.
> 1. Dicitionary-based approach for lemmatization has some limitations: the words out of the dictionary may incorrectly lemmatized.
>
>     **Response:** We have conducted an Out of Dictionary (OOD) test to measure the performance for words that are not present in the dictionary. The procedure encompassed the retrieval of recent data from diverse sources (e.g., newspaper, facebook comment, blogpost). Subsequently, we measured the performance of the lemmatizer on the OOD words only and found an accuracy of 79% where we used the BNLP automatic PoS tagger [1] to obtain the PoS tags. During analysis we found the following things:
>     - Though a word is not present in the dictionary, the inflection pattern is known for the particular word’s PoS class and sequential stripping of the markers leads to the correct lemma most of the time.
>
>     - The performance of the lemmatizer on OOD words can also increase upon using a more accurate or improved PoS tagger.
>
>     Thus, our method may produce incorrect results for OOD words, however, the sequential stripping proves to be an effective process to achieve generalizability for both OOD and dictionary words.
>
>      **References:**
>
>      [1] Sarker, Sagor. "Bnlp: Natural language processing toolkit for bengali language." arXiv preprint arXiv:2102.00405 (2021).
>
> 2. The authors evaluate their lemmatizer on the correct lemmas (only accuracy).
>
>     **Response:** We conducted additional experiments following your feedback to assess our lemmatizer's performance using our original test dataset. We evaluated its accuracy on two types of data: non-inflected words (these are usually the words found in the dictionary, for example exceptions may occur for proper nouns), inflected words (these words need to be lemmatized). In these considerations, there are 6906 non-inflected words and 3125 inflected words.
>
>      Our experimental results include a macro precision of 0.9537, macro recall of 0.9597, macro F1 score of 0.9567 and overall accuracy of 0.9636 (also reported in the paper) for these two categories of words. These findings will be included in the appendix of the final manuscript. We include the detail result for both the non-inflected and inflected words below:
>
>      **Non-inflected:**
>
>      - Precision : 0.9784245583550536
>
>      - Recall : 0.9681902851411377
>
>      - F1 Score : 0.9732805185451926
>
>      **Inflected:**
>
>      - Precision : 0.92896
>
>       - Recall : 0.9511795543905636
>
>       - F1 Score : 0.9399384814634937

---

### Official Review · Reviewer_N1LB · 2023-08-04

**Typos Grammar Style And Presentation Improvements:** 1. The word "processs" in the sentenc…
**Soundness:** 3

**Excitement:**

4: Strong: This paper deepens the understanding of some phenomenon or lowers the barriers to an existing research direction.

**Paper Topic And Main Contributions:**

The paper addresses the challenge of lemmatization for the Bangla language, an essential task in natural language processing (NLP) and linguistics.

Main contributions:

1. The paper presents BanLemma, a lemmatization system specifically tailored for Bangla. This system shows superior performance when compared to existing Bangla lemmatization methods.

2. The study provides a set of linguistic rules that explain how Bangla-inflected words are derived from their respective base words or lemmas. These rules result from a rigorous analysis of an expansive Bangla text corpus.

3. The linguistic rules are based on a comprehensive analysis of a Bangla text corpus of 90.65M unique sentences. This corpus includes 0.5B words, of which 6.17M are distinct. The study manually sampled and analyzed 22,131 words from this corpus to understand the formation of inflected words.

4. The paper establishes an evaluation framework, conducting intra-dataset and cross-dataset evaluations. BanLemma achieved an accuracy of 96.36% on intra-dataset testing and demonstrated robust performance across different datasets.

5. BanLemma outperformed several recently published methodologies in cross-dataset evaluations, indicating performance improvements ranging from 1% to 11%.

**Questions For The Authors:**

A. The paper mentions a unique approach of analyzing the suffix marker occurrence according to the morpho-syntactic values and then utilizing sequences of suffix markers instead of entire suffixes. How does this method differ from traditional methods, and what specific advantages does it offer for Bangla language processing?

B. The reported accuracy of BanLemma on your own human-annotated test set is 96.36%, which is quite impressive. However, there's a discrepancy in performance when using the test dataset provided by Chakrabarty et al. (2017). What attributes of their dataset or your lemmatizer could contribute to this discrepancy?

C. The methodology section highlights the importance of PoS tags for the lemmatizer to operate accurately. Given the dependency on the accuracy of the PoS tagger, are there plans to improve the PoS tagger or to incorporate multiple PoS tagging mechanisms to increase the robustness of the lemmatizer?

D. The limitations section identifies issues with out-of-dictionary words. Considering this limitation, how does the BanLemma handle real-world scenarios where the system might frequently encounter such words, and what are the proposed strategies to mitigate this limitation in future iterations?

**Reasons To Accept:**

Strengths:

1. The paper introduces BanLemma, based on a detailed and rigorous analysis of a large Bangla text corpus, allowing for a comprehensive understanding of Bangla word formation.

2. Instead of focusing on entire suffixes, BanLemma analyzes suffix marker sequences, addressing the limitation of previous lemmatization methods.

3. Achieving an accuracy of 96.36% on a human-annotated test dataset is commendable and shows the potential and efficiency of BanLemma.

4. The paper demonstrates the ability of BanLemma to perform competitively on various datasets, surpassing other lemmatizers on certain test sets.

5. The paper delves deeply into its methodology, detailing the creation of rules and dictionary for BanLemma, and the extensive sources they utilized.

Benefits to the NLP community:

1. The paper tackles lemmatization in Bangla, a less-represented language in NLP. As Bangla is morphologically rich, addressing lemmatization in this context is essential for many higher-level NLP tasks.

2. The comprehensive dataset, rules, and methods introduced can be a foundational resource for future researchers working on Bangla NLP.

3. By combining linguistic rules with NLP techniques, the paper demonstrates the potential benefits of such hybrid methods, possibly encouraging more linguistically-informed approaches in the field.

**Reasons To Reject:**

Weaknesses:

1. The paper notes a significant decrease in performance when used with an automatic PoS tagger. A lemmatizer that relies heavily on the accuracy of an external PoS tagger may suffer in real-world applications if the PoS tags aren't accurate.

2. The described limitation wherein the lemmatizer strips suffixes from words that are already lemmas but end with a suffix substring is a major concern for real-world texts with evolving vocabularies.

3. The lemmatizer demonstrates notably lower performance on the dataset provided by Chakrabarty et al. (2017). Such variance in performance on different datasets indicates potential overfitting to certain data types or lack of generalizability.

4. While the paper mentions that the rules were derived from rigorous analysis, the specifics of the rules and their formation aren't deeply discussed. This makes it hard to gauge the lemmatizer's adaptability to nuanced linguistic challenges.

5. The examples where the lemmatizer fails are linked to the PoS tagger's inaccuracies. A more comprehensive analysis of the failure cases, beyond PoS errors, would have provided a holistic view of the lemmatizer's weaknesses.

Risks:

1. If presented at a conference, attendees may question the lemmatizer's generalizability, given its varied performance on different datasets.

2. The heavy dependency on PoS taggers could raise questions about the stand-alone efficacy of BanLemma. If other researchers attempt to reproduce results or integrate the lemmatizer into different systems, the variance in PoS tagging quality may produce inconsistent results.

3. Without clear details on rule formation and limitations, other researchers may misapply or misinterpret the lemmatizer's capabilities, leading to unintended consequences in subsequent research or applications.

**Reproducibility:**

3: Could reproduce the results with some difficulty. The settings of parameters are underspecified or subjectively determined; the training/evaluation data are not widely available.

**Reviewer Confidence:**

3: Pretty sure, but there's a chance I missed something. Although I have a good feel for this area in general, I did not carefully check the paper's details, e.g., the math, experimental design, or novelty.

---

> ### Author Rebuttal · Authors · 2023-08-28
>
> Thank you for your valuable feedback, and we believe these will improve the quality of the paper. We are writing our response to each point separately below.
>
> **Our response to the ”Questions For The Authors":**
> 1. The paper mentions a unique approach of analyzing the suffix marker occurrence according to the morpho-syntactic values and then utilizing sequences of suffix markers instead of entire suffixes. How does this method differ from traditional methods, and what specific advantages does it offer for Bangla language processing?
>
>     **Response:** Below we explain how the proposed method differs including its specific advantages over other existing methods:
>      - The traditional method of stripping an entire suffix fails to cover the lemmatization of different Bangla word formations because of its highly inflected nature. Bangla words are inflected with four different markers (e.g., Case marker, Plural marker, Determiner, and Emphasis marker) that convey grammatical meanings and there are some specific sequences in their occurrence [1]. Moreover, the structure of Bangla words supports using multiple suffixes sequentially as well as with different combinations of the same word. To make it more challenging, the lemma may end with characters that are similar to the inflection markers [2]. This is why stripping the entire suffix fails even in the most common cases for Bangla lemmatization.
>
>           For example, ‘মেয়েরটা’ (meʲerta; girl’s) in this word lemma is মেয়ে (meʲe; girl) and inflected with র (r) and টা(ta). Another word মায়েরটা (maʲerta; mother’s) where মা (ma; mother)  is the lemma and inflected with য়ের (er) and টা (ta). Traditional lemmatizer would strip the য়েরটা (erta) in both cases resulting in invalid lemma মে (me) in মেয়েরটা (meʲerta) as the lemma ends with য়ে (ʲe). Here, BanLemma will strip টা (ta) first and র then cross-reference with the dictionary and return the accurate lemma মেয়ে (meʲe; girl). So stripping morpho-syntactic sequences of characters at a time produces better results than stripping entire suffixes.
>
>
>
>      - BanLemma covers almost all possible suffixes by using the combination of markers rather than just a list of fixed suffixes (No previous work found that could cover all the suffixes). Moreover, we will also release our suffix list upon acceptance of the paper.
>      - As markers are limited and generally very short in length, often 1-3 characters, it provides computational advantages rather than matching a long list of suffix strings.
>
>     **References:**
>
>      [1] Bhattacharya, Samit, et al. "Inflectional morphology synthesis for bengali noun, pronoun and verb systems." Proc. of the National Conference on Computer Processing of Bangla (NCCPB 05). 2005.
>
>      [2] Islam, Rafiqul & Sarkar, Pabitra (2017). Bangla Academy Pramita Bangla Bhashar Byakaran. Dhaka: Bangla Academy.
>
> 2. The reported accuracy of BanLemma on your own human-annotated test set is 96.36%, which is quite impressive. However, there's a discrepancy in performance when using the test dataset provided by Chakrabarty et al. (2017). What attributes of their dataset or your lemmatizer could contribute to this discrepancy?
>
>     **Response:** We have observed several discrepancies in Chakrabarty et al. (2017) efforts of annotating lemmatizing Bangla words. We have discussed the type of errors (to the best of our knowledge) in their work below:
>
>      - Chakrabarty et al. (2017) made changes to the gender of a set of words of their choice, such as transforming যুবতী (ɟubot̪i; young girl) to যুবক (ɟubok; young boy). However, according to [2] lemmatization does not intend to change the semantic meaning of a word.
>
>     - As discussed in [1], inflection does not involve adding prefixes and altering the word's meaning. But in their dataset, by stripping prefixes they altered negated forms to positive forms, such as changing অদূর (ɔd̪ur; not so far) to দূর (d̪ur; far).
>
>     - They kept inconsistent annotations for pronouns, such as তোমার (your) is annotated as both তুমি (t̪umi; you) and তোমার (t̪omar; your). We always considered the lemma of তোমার (t̪omar; your) as তোমার (t̪omar; your).
>
>      - They made derivational changes, such as transforming বাণিজ্যিক (baniɟik; commercial) to বাণিজ্য (baniɟɟo; trade), প্রকৃতি (prokrit̪i; nature)  to প্রকৃত (prokrit̪o; real), and so on. But in study [2] authors discuss that lemmatizer does not involve derivational changes and we also do the same in BanLemma.
>      - They had errors, including misspelling and incorrect lemmatization of words such as transforming সংশোধনাগার (ʃɔŋʃod̪ʱonagar; reformatory) as সংশোধনঅগার (ʃɔŋʃod̪ʱonɔgar; )
>
>     To effectively evaluate our proposed system, we made an attempt to check and correct some erroneous lemmas. We took 52 sentences at random which contained about 695 words and corrected the lemma as well as annotated the PoS tags. Upon acceptance, we plan to open the sampled portion along with the corrected lemma and annotated PoS tags. On this sampled portion, we achieved an accuracy of 94.34% measured against the corrected lemmas and using the annotated PoS tags. This performance is reported in Table 7 of the paper.
>
>     **References:**
>
>      [1] Lieber, Rochelle. Introducing morphology. Cambridge University Press, 2021.
>
>      [2] Karwatowski, Michal, and Marcin Pietron. "Context based lemmatizer for Polish language." arXiv preprint arXiv:2207.11565 (2022).
>
>
> 3. The methodology section highlights the importance of PoS tags for the lemmatizer to operate accurately. Given the dependency on the accuracy of the PoS tagger, are there plans to improve the PoS tagger or to incorporate multiple PoS tagging mechanisms to increase the robustness of the lemmatizer?
>
>     **Response:**
>     Lemmatization considers a word's part of speech (PoS) in order to accurately ascertain its lemma, given that the lemma's meaning can undergo modifications based on its grammatical role [1] [2]. Our exploration of prior lemmatization research revealed that prominent contemporary methodologies, such as Trankit [3], TurkuNLP [4], and LEMMING [5], utilize PoS tags before the lemmatization process. Drawing inspiration from these valuable findings, we purposefully tailored our approach to align with this prevailing paradigm, particularly due to our involvement in an expansive initiative dedicated to advancing Bangla NLP research and its practical implementations. Concurrently, within this overarching framework, we are actively involved in enhancing the automated PoS tagging process. Notably, the present study is exclusively centered on refining the lemmatization methodology for the Bangla Language.
>
>     **References:**
>
>      [1] Bhattacharya, Samit, et al. "Inflectional morphology synthesis for bengali noun, pronoun and verb systems." Proc. of the National Conference on Computer Processing of Bangla (NCCPB 05). 2005.
>
>      [2] Karwatowski, Michal, and Marcin Pietron. "Context based lemmatizer for Polish language." arXiv preprint arXiv:2207.11565 (2022).
>
>      [3] Minh Van Nguyen, Viet Dac Lai, Amir Pouran Ben Veyseh, and Thien Huu Nguyen. 2021. Trankit: A Light-Weight Transformer-based Toolkit for Multilingual Natural Language Processing. In Proceedings of the 16th Conference of the European Chapter of the Association for Computational Linguistics: System Demonstrations, pages 80–90, Online. Association for Computational Linguistics.
>
>      [4] Kanerva, Jenna, et al. “Universal Lemmatizer: A Sequence-to-Sequence Model for Lemmatizing Universal Dependencies Treebanks.” Natural Language Engineering, vol. 27, no. 5, 2021, pp. 545–574., doi:10.1017/S1351324920000224.
>
>      [5] Müller, Thomas, et al. "Joint lemmatization and morphological tagging with lemming." Proceedings of the 2015 conference on empirical methods in natural language processing. 2015.
>
> 4. The limitations section identifies issues with out-of-dictionary words. Considering this limitation, how does the BanLemma handle real-world scenarios where the system might frequently encounter such words, and what are the proposed strategies to mitigate this limitation in future iterations?
>
>     **Response:**
> 	Thank you for your concern. In response to your query, an assessment of the generalizability of the proposed method was conducted through an out-of-dictionary (OOD) test. The procedure encompassed the retrieval of recent data from diverse sources (e.g., newspaper, facebook comment, blogpost). Subsequently, we measured the performance of the lemmatizer on the out-of-dictionary words only and found an accuracy of 79% where we used the BNLP automatic PoS tagger [1] to obtain the PoS tags. During analysis we found the following things:
>
>       - Though a word is not present in the dictionary, the inflection pattern is known for the particular word’s PoS class and sequential stripping of the markers leads to the correct lemma most of the time.
>
>       - The performance of the lemmatizer on OOD words can also increase upon using a more accurate or improved PoS tagger.
>
> 	Other than improving the PoS tagger, we plan to take the following steps:
>
>       - Enriching the dictionary: Using the vast data available in different online sources, we would take steps to identify keywords and phrases and enrich the dictionary.
>
>       - Hybrid system: Leveraging this system we plan to create a Bangla lemmatization data set and develop a neural network based or hybrid lemmatization system in future.
>
>      **References:**
>
>       [1] Sarker, Sagor. "Bnlp: Natural language processing toolkit for Bengali language." arXiv preprint arXiv:2102.00405 (2021).
>
>
> **Our response to the “Reasons To Reject (Weakness)":**
> 1. The paper notes a significant decrease in performance when used with an automatic PoS tagger. A lemmatizer that relies heavily on the accuracy of an external PoS tagger may suffer in real-world applications if the PoS tags aren't accurate.
>
>     **Response:** Kindly refer to question 3 in "Questions to Authors" section, where we have discussed the dependency of PoS tagger and outlined our strategies for addressing them in upcoming iterations.
> 2. The described limitation where in the lemmatizer strips suffixes from words that are already lemmas but end with a suffix substring is a major concern for real-world texts with evolving vocabularies.
>
>     **Response:** Kindly refer to question 3 in "Questions to Authors" section, where we have discussed the case regarding our lemmatizer performance on Out of Dictionary (OOD) words.
> 3. The lemmatizer demonstrates notably lower performance on the dataset provided by Chakrabarty et al. (2017). Such variance in performance on different datasets indicates potential overfitting to certain data types or lack of generalizability.
>
>     **Response:** In question 2 of “Questions to Authors” we have discussed the case performance discrepancy on Chakrabarty et al. (2017) dataset.
> 4. While the paper mentions that the rules were derived from rigorous analysis, the specifics of the rules and their formation aren't deeply discussed. This makes it hard to gauge the lemmatizer's adaptability to nuanced linguistic challenges.
>
>     **Response:** Thank you for your feedback. We have already discussed our proposed rules in detail with their formation in our article (please see section 3.1 and appendix A.3). However we have a plan to write up an in-detailed document and release it to the public through GitHub along with code upon acceptance.
> 5. The examples where the lemmatizer fails are linked to the PoS tagger's inaccuracies. A more comprehensive analysis of the failure cases, beyond PoS errors, would have provided a holistic view of the lemmatizer's weaknesses.
>
>     **Response:** In the limitations section of the paper, we covered instances where the lemmatizer encounters challenges, encompassing scenarios like Out of Dictionary words, words with ambiguous semantic meanings etc.
>
>
> **Our response to the “Reasons To Reject (Risks)":**
>
> 1. If presented at a conference, attendees may question the lemmatizer's generalizability, given its varied performance on different datasets.
>
>     **Response:** Thank you for your valuable response. In our experimentation we have taken utmost care for achieveing generalizability of our proposed BanLemma. From our in-depth experiments, we have observed that, the baseline datasets we use has several grammatical errors. Below we will mention the key errors we have found from our experimentations, however the detail analysis can be found in our submitted article (Section: Appendix (A.9), from line 930).
>
>     - BenLem dataset: Labeled verb roots differently from dictionary forms; converted colloquial to classical pronouns; spelling changes impacted lemmatization.
>
>     - Banel: Inconsistent derivational marker removal; altered pronoun forms; spelling changes affected outcomes; contained incorrect lemmas.
>
>     - Chakrabarty et al. (2017): Modified gender, negated forms, and pronouns; applied derivational changes impacting lemmatization, erroneous PoS labeling.
>
>     It is worth noting that, inspite of this error, we have achieved competitive results on BaNeL and BenLem dataset. As we have mentioned earlier in terms of data discrepancies, we have randomly choose 52 sentences (more than 700 words) from that dataset and corrected their mistakes before testing with our proposed method. By correcting the random samples from Chakrabarty et al. (2017) we have gained a 3% performance improvement. The aforementioned statement has already been mentioned in the Results section (See table 6, 7 and line 550-572).
>
>      **References:**
>
>      [1] Islam, M.A., Towhiduzzaman, M., Bhuiyan, M.T.I. et al. BaNeL: an encoder-decoder based Bangla neural lemmatizer. SN Appl. Sci. 4, 138 (2022). https://doi.org/10.1007/s42452-022-04985-2
>
>
> 2. The heavy dependency on PoS taggers could raise questions about the stand-alone efficacy of BanLemma. If other researchers attempt to reproduce results or integrate the lemmatizer into different systems, the variance in PoS tagging quality may produce inconsistent results.
>
>     **Response:** We would like to thank you for your feedback. The use of parts of speech lemmatization has proven to be very useful in achieving state-of-the-art results (e.g., Trankit [1], TurkuNLP [2], and LEMMING [3]). Thus we believe that reproducing the results should not have any problem since we have used publicly available resources such as BNLP PoS tagger [1] which is available online to ensure that we can avoid inconsistency in our results reported in this study. Additionally, we will publish our code, dictionary, and all other resources related to this study through GitHub upon acceptance of this paper.
>
>      **References:**
>
>      [1] Minh Van Nguyen, Viet Dac Lai, Amir Pouran Ben Veyseh, and Thien Huu Nguyen. 2021. Trankit: A Light-Weight Transformer-based Toolkit for Multilingual Natural Language Processing. In Proceedings of the 16th Conference of the European Chapter of the Association for Computational Linguistics: System Demonstrations, pages 80–90, Online. Association for Computational Linguistics.
>
>     [2] Kanerva, Jenna, et al. “Universal Lemmatizer: A Sequence-to-Sequence Model for Lemmatizing Universal Dependencies Treebanks.” Natural Language Engineering, vol. 27, no. 5, 2021, pp. 545–574., doi:10.1017/S1351324920000224.
>
>      [3] Müller, Thomas, et al. "Joint lemmatization and morphological tagging with lemming." Proceedings of the 2015 conference on empirical methods in natural language processing. 2015.
>
>
> 3. Without clear details on rule formation and limitations, other researchers may misapply or misinterpret the lemmatizer's capabilities, leading to unintended consequences in subsequent research or applications.
>
>     **Response:** We have made an effort to provide a comprehensive explanation of our rules in the manuscript (see section 3 and Appendix A.3). With due respect to your feedback, we will create elaborate descriptions of each lemmatizing rule and release those through GitHub. We will also open source all our code related to this study upon acceptance of this paper.
>
>
>
> **Regarding Typos Grammar Style And Presentation Improvements:**
> We sincerely apologize for the typographical errors. We will take the necessary steps to revise the paper and ensure that all mistakes are rectified.

---

### Official Review · Reviewer_5hHh · 2023-08-06

**Soundness:** 4

**Excitement:**

4: Strong: This paper deepens the understanding of some phenomenon or lowers the barriers to an existing research direction.

**Paper Topic And Main Contributions:**

This paper presents a rule-based approach for predicting the lemma for one of the highly inflected languages; the Bangala language. Nevertheless, it is still one of the low-resource languages. For predicting the lemma of each word, they mainly depend on it Part-of-Speech (POS) where each POS has its rules that strip the suffixes in different ways. In addition, they utilize a look-up dictionary to check the presence of the predicted lemma after striping the suffixes. They analyze the suffix marker occurrence according to the morpho-syntactic values and then utilized sequences of suffix markers instead of entire suffixes. For developing their rules, they analyze a large corpus of Bangla text from various domains, sources, and time periods to observe the word formation of inflected words. They achieve SOTA results when tested against a manually annotated test dataset against the three previously published Bangla lemmatization datasets.

**Reasons To Accept:**

The authors of this paper present a rule-based lemmatization system for one of the low-resource and rich morphology languages. They provide a detailed linguistic description of how they developed the rules depending, mainly, on the POS of each word where the POS is the key for the suffix markers to be stripped from the word and, then, check their developed dictionary. In addition, they analyzed a large corpus of Bangala texts that covers different sources and domains. They compare their work with the three previously published trials for predicting the lemma and they could achieve SOTA results against these systems. They achieve an accuracy of 96.67% and list the limitations of their system that they may work on it, in the future, for enhancing the results of their lemmatizer.

**Reasons To Reject:**

No reasons

**Reproducibility:**

4: Could mostly reproduce the results, but there may be some variation because of sample variance or minor variations in their interpretation of the protocol or method.

**Reviewer Confidence:**

4: Quite sure. I tried to check the important points carefully. It's unlikely, though conceivable, that I missed something that should affect my ratings.

**Typos Grammar Style And Presentation Improvements:**

The paper is well-written and easy to read, nevertheless, I have some comments:

1- The writing style of Bangala words with their English pronunciation is inconsistent. It is recommended to use a unified style, for example, (Bangala word + it transcription in IPA format + English meaning). This would help researchers of other languages to read the paper easily.

2- There are some typos:
       A. Line 114: tas.
       B. Line 174: lemmatizat.
       C. Line 866: a letter is written as a square.
       D. Line 879: don't add comma separators in numbers since it conflicts with the comma of the written sentence itself.

---

> ### Author Rebuttal · Authors · 2023-08-28
>
> Thanks for your thoughtful comment and kind words. We'll respond to each point separately below:
> 1. The writing style of Bangala words with their English pronunciation is inconsistent. It is recommended to use a unified style, for example, (Bangala word + it transcription in IPA format + English meaning). This would help researchers of other languages to read the paper easily.
>
>    **Response:** Thank you for providing your valuable suggestions. We will ensure that they
>        	are incorporated into the revised manuscript.
> 2. There are some typos: A. Line 114: tas. B. Line 174: lemmatizat. C. Line 866: a letter is written as a square. D. Line 879: don't add comma separators in numbers since it conflicts with the comma of the written sentence itself.
>
>    **Response:** We sincerely apologize for the typographical errors. We will take the necessary steps to revise the paper and ensure that all mistakes are rectified.

---

### Meta-Review · Area_Chair_ajBe · 2023-09-11

**Recommendation:** 4

**Metareview:**

This paper presents a rule-based Bangla lemmatizer. It strips suffix markers and validates the predicted lemma against a dictionary.

It is somewhat unexpected to encounter a rule-based language analyzer in 2023. However, in comparison to prior research, this paper demonstrates relatively strong performance. The authors conducted comprehensive experiments and transparently outlined the study's limitations. Although it may not reach a broad audience, this paper deserves publication in some form.

---

### Decision · Program_Chairs · 2023-10-07

**Decision:**

Accept-Findings

**Comment:**

This paper presents a rule-based Bangla lemmatizer. It strips suffix markers and validates the predicted lemma against a dictionary.

It is somewhat unexpected to encounter a rule-based language analyzer in 2023. However, in comparison to prior research, this paper demonstrates relatively strong performance. The authors conducted comprehensive experiments and transparently outlined the study's limitations. Although it may not reach a broad audience, this paper deserves publication in some form.